# Modal Planning for Cooperative Non-Prehensile Manipulation by Mobile Robots

**Changxiang Fan** [1,*], **Shouhei Shirafuji** [2] **and Jun Ota** [2]

1   Department of Precision Engineering, Graduate School of Engineering, The University of Tokyo, 7-3-1 Hongo, Bunkyo-ku, Tokyo 113-8656, Japan
2   Research into Artifacts, Center for Engineering, The University of Tokyo, 5-1-5 Kashiwanoha, Kashiwa-shi, Chiba 277-8568, Japan; shirafuji@race.u-tokyo.ac.jp (S.S.); ota@race.u-tokyo.ac.jp (J.O.)
*   Correspondence: fan@race.u-tokyo.ac.jp; Tel.: +81-04-7136-4276

**Abstract:** If we define a mode as a set of specific configurations that hold the same constraint, and if we investigate their transitions beforehand, we can efficiently probe the configuration space by using a manipulation planner. However, when multiple mobile robots together manipulate an object by using the non-prehensile method, the candidates for the modes and their transitions become enormous because of the numerous contacts among the object, the environment, and the robots. In some cases, the constraints on the object, which include a combination of robot contacts and environmental contacts, are incapable of guaranteeing the object's stability. Furthermore, some transitions cannot appear because of geometrical and functional restrictions of the robots. Therefore, in this paper, we propose a method to narrow down the possible modes and transitions between modes by excluding the impossible modes and transitions from the viewpoint of statics, kinematics, and geometry. We first generated modes that described an object's contact set from the robots and the environment while ignoring their exact configurations. Each multi-contact set exerted by the robots and the environment satisfied the condition necessary for the force closure on the object along with gravity. Second, we listed every possible transition between the modes by determining whether or not the given robot could actively change the contacts with geometrical feasibility. Finally, we performed two simulations to validate our method on specific manipulation tasks. Our method can be used in various cases of non-prehensile manipulations by using mobile robots. The mode transition graph generated by our method was used to efficiently sequence the manipulation actions before deciding the detailed configuration planning.

**Keywords:** non-prehensile manipulation; manipulation planning; contact planning; manipulation action sequences

## 1. Introduction

Spatial restrictions make it almost impossible to manipulate a big object in a narrow space by using big-scaled manipulators. For instance, it is impractical to carry an industrial manipulator into our house to move furniture by grasping and lifting. Owing to their small size and flexibility in motion, multiple mobile robots can be adopted to perform tasks in a narrow space [1]. These robots can move in a narrow environment to approach and manipulate objects, but these robots cannot grasp big objects as large-scale industrial manipulators. Therefore, non-prehensile methods [2,3], which involves manipulation without grasping, is practical for such cases. For instance, a preferred way to manipulate a big object is to push it along the floor, or to pivot it with a vertex that makes contact with the floor. In certain cases, the object keeps contacting the environment in such manipulations and the restrictions on the object motion caused by the contacts complicates the kinematics in the manipulation.

Furthermore, when multiple mobile robots perform a manipulation task, they themselves form a complex coordinated system [4]. Consequently, non-prehensile manipulations that use multiple mobile robots (as shown in Figure 1) require convoluted manipulation planning than standard manipulation, which comprehensively takes into account each robot's kinematics, the surrounding constraints, and their changes.

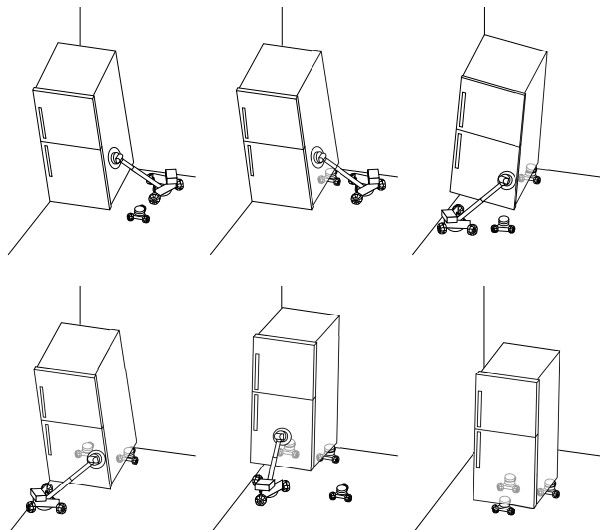

**Figure 1.** Example of non-prehensile transportation adopting multiple mobile robot: Two types of mobile robots move the refrigerator placed at the corner of a room.

Multiple robot motion planning is often faced with the high dimensional configuration space [5]. Typical planners for such problems are sample-based, such as RRT [6] and PRM [7]. However, manipulation planning problems often encounter particular *multi-modal* structures [8], if the contacts among the robots, the manipulated objects, and the environment change during the process of manipulation. Here, a *mode* refers to a certain set of configurations that hold the same constraints in motion (e.g., the object motion keeping a set of contacts with the environment). Possible configurations under the same constraints form sub-spaces in the configuration space. This requires the planner to be capable not only to probe the sub-spaces of each mode but also to cross among the different sub-spaces. For the application of typical sample-based methods on their original spaces, the expansiveness among the configurations of the different modes is more difficult to achieve than those of the same mode [9]. For instance, we can generate a transitable configuration of the system for a current configuration by sampling the configuration the belongs to the sub-space, while keeping the contact set among the robots, the objects, and the environment unchanged. However, in case the contact set changes, it would be necessary to check the connectivity between the different sub-spaces corresponding to the contact states, which would complicate the problem.

Therefore, typical sample-based methods are usually applied to the modes' sub-spaces after splitting the configuration space into sub-spaces according to the modes. The sample-based methods become realizable by deciding the sequence of modes where connectivity is guaranteed beforehand. Maeda et al. [10] and Miyazawa et al. [11] sampled the manipulation states in a configuration space where a sequence of modes existed and the modes' transitions were prior defined. Some planners have been proposed for creating the modes' roadmap to guide the manipulation sequencing [8,12–14]. In particular, Lee [13] adopted a PRM-based planner [15,16] to split the modes by comprehensively considering the multi-contacts between the object, robots, and the environment to obtain the necessary modes to pass through for a manipulation task. Mode transitions are mainly derived from the compliant transformations of contacts between the object, the environment, and the contacts of

the given robots; two robot contacts in the examples [13] were adopted to realize the subsequent mode transitions.

However, when multiple mobile robots manipulated objects by using non-prehensile methods, mode transitions became more complex, because besides considering the mode transitions caused by the changes in the environmental contact, robot contacts also had to be considered. One way of addressing this complexity is by eliminating unfeasible modes in statics from the enormous combinations of contacts among them before considering the transitions among modes. An object should be under sufficient constraints in each mode; otherwise, the robots will fail to manipulate the object (e.g., the object drops down in an unexpected direction because of the lack of constraints). This means that the contacts from the environment and the robots should be able to form full constraints on the objects for non-prehensile manipulations. If we consider the environmental contact alone, the resultant constraints would vary in different contact states. For example, when an object–floor contact state changes from face–face contact into vertex–face contact, the constraints exerted by the floor would reduce. This results in various least requirements of robot constraints under different environmental constraints. Sometimes, we require the object to keep stationary contact with the environment, so that the robot can use friction to move the object (e.g., inclining a box); sometimes, we require the robots to manipulate the object to slide along the contacting part (e.g., sliding a box on a floor). To distinguish these manipulations, the environmental contact should be identified into the fixed contact and the sliding contact. The restraint placed on the object's degrees of freedom by a contact is different between the cases when it is fixed or sliding. Thus, we consider the necessary amount of robot for manipulations, both when relative sliding happened and did not happen. Therefore, a proper consideration of the individual robot's kinematics and the changes in the environmental constraints is essential when splitting the modes.

Furthermore, a robot contact changes when the robot makes or breaks contact with the object either actively or passively (by the object's motion). For example, in Figure 1, a large robot with a manipulator actively makes contact with the object and pushes it over the small robots, and the small robots passively make contact with the object by the action of the large robot. The distinction between the active and passive action appears in many manipulation tasks, and the possible sequence of actions depends on this distinction of actions. Thus, the planner should be able to reason about all such possible mode transitions.

By addressing the problems peculiar to non-prehensile manipulations using the mobile robots described above, we propose a method to generate the modes and transitions between the modes for contact planning. In our method, for a given set of contacts, we identified the modes by analyzing its constraints and least requirements to constrain the object's motion. We determine the mode transitions based on how the robot contact influenced the contact state of a targeted object. The mobile robots were divided into active robots and passive robots according to how their contacts changed in the state transition. Finally, the manipulation actions were determined by sequencing a series of modes. We applied the same concept to the limited cases also [17]. In this paper, we propose a more generalized framework to determine the action sequence including the distinction between the fixed and the sliding contact states.

In the second section, we describe the problem statement, and in the third section, we introduce the generation of the contact state. In the fourth section, the state transition is investigated to sequence the action series. In the fifth section, we describe the applications of our methodology by conducting two simulations of the specified manipulation cases.

## 2. Problem Statement

In this paper, for simplicity, we have only manipulated objects having convex polyhedrons. Furthermore, we consider that the objects, the environment, and the robots are all rigid. A set of contacts on an object having environmental contact and robotic contact were represented as a contact

state and defined as a mode. A contact state is described without the exact position of contact and configuration of the object and the robots.

For a description of the contact state without the exact configurations, we adopt the concept of principal contact (PC) [18] to express the contact states. A PC is a contacting pair between the geometrical primitives (a vertex, an edge, or a face). A PC is denoted by $c = (a, b)$, where $a$ is a geometrical primitive on the object and $b$ is a geometrical primitive on a polyhedron of the environment or a robot. Furthermore, we denote the object–environment PC and object–robot PC by $c^{\mathrm{e}}$ and $c^{\mathrm{r}}$, respectively, for clarity.

In our method, we need to distinguish whether relative sliding happening on a contact for determining the constraints required to realize the full constraints of the object. For example, Figure 2 shows the difference of the required constraints in the quasi-static manipulation of an object that lies on a floor with an edge contact when a robot contacts with it as a point–face contact (in the planar case). Manipulation without relative sliding on the point with the help of floor's static friction, as shown in Figure 2c, requires a robot to tilt the object. However, for manipulation with relative sliding between the object and the floor, it is difficult to predict the resultant motion of the object because of dynamic friction, as shown in Figure 2b. To guarantee that the target motion will be realized, an additional robot is required (see Figure 2c), and the number of robots required is different between the two cases.

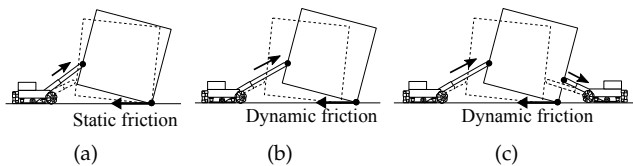

Static friction  Dynamic friction  Dynamic friction

(a)  (b)  (c)

**Figure 2.** Frictional constraint on object's motion in fixed and sliding cases: (**a**) A robot tilts an object with fixed contact on the floor; (**b**) a robot tilts an object with sliding contact; (**c**) two robots tilts an object with sliding contact.

Therefore, we distinguish whether relative sliding happens or not on a contact. A non-sliding PC is defined as *static* PC, denoted by $c^{\mathrm{e,st}} = (a, b)^{\mathrm{e,st}}$. Correspondingly, a sliding PC is defined as *dynamic* PC, denoted by $c^{\mathrm{e,dn}} = (a, b)^{\mathrm{e,dn}}$.

When multiple robots manipulate objects, not all of them actively make contact with an object. A contact occurs either when the robot actively touches the object or passively touches it when the object is moved. Furthermore, some robots move an object actively and change its state, whereas some robots operate as auxiliaries in the manipulation task. The distinction between the active and passive functions of a contact is important for considering the possible transitions of the states. Accordingly, the contacts are divided into active and passive and are denoted as $c^{\mathrm{r,st,ac}}$ and $c^{\mathrm{r,st,ps}}$, respectively. A robot with active joints can also act as a passive robot. Whether the robot always acts passively or actively or is switchable between passive and active depends on the function of the robot. A contact between an object and a robot usually does not slide. Therefore, for simplicity, we omit the subscript for the static and the dynamic contacts if the contact is between an object and a robot and write as $c^{\mathrm{r,ac}}$ and $c^{\mathrm{r,ps}}$. However, a contact between an object and the environment is passive. Therefore, we omit the subscript for the passive and the active contacts if the contact is between an object and the environment and write as $c^{\mathrm{e,st}}$ and $c^{\mathrm{e,dn}}$.

The set of PCs on an object is called contact formation (CF) [18] and denoted by $C$. To describe the original CF proposed by Xiao and Zhang [18], which does not concern the sliding of the contact point, we call a CF without distinguishing the static and dynamic as the *primitive* contact states, and denote it as $\widehat{C}$. For example, $\{(a_1, b_1)^{\mathrm{e}}, (a_2, b_2)^{\mathrm{e}}\}$ is the primitive state of $\{(a_1, b_1)^{\mathrm{e,st}}, (a_2, b_2)^{\mathrm{e,st}}\}$ or $\{(a_1, b_1)^{\mathrm{e,st}}, (a_2, b_2)^{\mathrm{e,dn}}\}$. Furthermore, when CFs are the same from the viewpoint of the primitive contact states, we describe them as *isogenous*. For example, $\{(a, b)^{\mathrm{e,st}}\}$ and $\{(a, b)^{\mathrm{e,dn}}\}$ are isogenous, and $\{(a_1, b_1)^{\mathrm{e,st}}, (a_2, b_2)^{\mathrm{e,st}}\}$ and $\{(a_1, b_1)^{\mathrm{e,st}}, (a_2, b_2)^{\mathrm{e,dn}}\}$ are also isogenous.

In this paper, we distinguish the CFs consisting of the contact states between an object and the environment, which we call environmental contact formation (ECF). We also distinguish the CFs consisting of the contact states between the robots and the environment, which we call robot contact formation (RCF); see Figure 3.

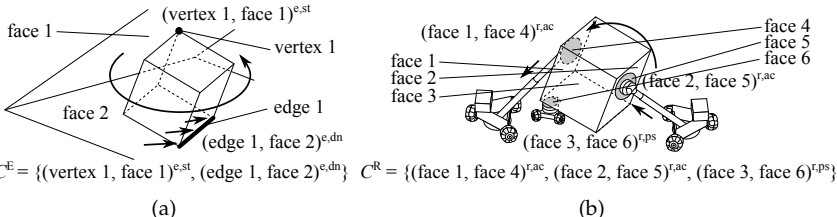

$C^E = \{(\text{vertex 1, face 1})^{\text{e,st}}, (\text{edge 1, face 2})^{\text{e,dn}}\}$   $C^R = \{(\text{face 1, face 4})^{\text{r,ac}}, (\text{face 2, face 5})^{\text{r,ac}}, (\text{face 3, face 6})^{\text{r,ps}}\}$

(a)                                                                 (b)

**Figure 3.** Examples of contact formation (CF): (**a**) environmental contact formation (ECF) when a cuboid contacts with two surfaces of the environment by its vertex and edge; (**b**) robot contact formation (RCF) when a cuboid contacts with two active robots and a passive robot by its faces.

Furthermore, we describe them as $C^E$ and $C^R$, respectively. As the result, we denote the mode $s$ in the manipulation as the set of ECF and RCF:

$$s = \{C^E, C^R\}.$$

A multi-contact set on an object was supposed to form the full constraints, so that the robots manipulated the object quasi-statically. Gravity closure [19,20], which is force closure that includes the gravitational force, is introduced later in this paper as the requirement for a mode. Our first goal is to identify all possible modes from the viewpoint of gravity closure.

If one mode can directly transform into another mode without any intermediate ones, it is a possible mode transition. Mode transitions can be described by a graph that comprises nodes and arcs, where the nodes represent individual modes, and the arcs between them represent the transitions; the value of an individual arc represented the cost of the state transition [21]. Our second goal is to generate this mode graph by taking into account some restrictions on robots and the geometrical relationships between an object and the environment. Using the resultant graph, the manipulation action sequences are determined by searching for paths from a given initial mode to a targeted mode before applying a sample-based method to probe the configuration spaces determined by modes.

In the following sections, we have made the following assumptions. The shape and the gravity center of the object and the shape of the environment are given. Furthermore, we have assumed that the type of contact that a robot generates with the object is given, and it does not change in the manipulation. The number of robots is also given. We have considered only the possible RCF on the targeted object, and the contacts between the robots and the environment were ignored; the robots generally make contact with a face (a floor) in the environment.

## 3. Generation of Modes

In this section, we identified all possible modes from the viewpoint of gravity closure. Given a set of mobile robots and an object lying in a certain environment, we obtained the possible modes using the following steps: (i) The possible ECFs were specially identified. We identified the ECFs before identifying the RCFs because the ECF restrains the robot's accessible area. (ii) For the identified ECF, we identified the possible RCF. (iii) Finally, by combining the ECFs and the RCFs, full constraints could be achieved.

### 3.1. Generation of ECFs

The geometrical relationship between an object's shape and the shape of its environment determines the possible ECFs. A PRM-based sampling strategy has been proposed to investigate

the possible CFs between objects [15,16,22]. In their method, the target object's orientation was incrementally changed with collisions checked to obtain the possible CFs. If a new CF appeared, it was recorded on the list, along with the object's current orientation. In this way, all the possible CFs between the objects could be probed; each CF was recorded with an available object orientation. We adopted this method to investigate the possible ECFs in our planner.

As mentioned in the previous section, we distinguished the contacts into static and dynamic contact based on the resultant CFs obtained by their method. For the given CFs obtained by their method, every PC was split into static and dynamic PCs, and ECFs are given as all combinations of the split static and dynamic PCs. For example, given a CF $C = \{c_1, c_2\}$, the ECFs are $C_1^E = \{c_1^{e,st}, c_2^{e,st}\}$, $C_2^E = \{c_1^{e,st}, c_2^{e,dn}\}$, $C_3^E = \{c_1^{e,dn}, c_2^{e,st}\}$, and $C_4^E = \{c_1^{e,dn}, c_2^{e,dn}\}$.

Using the above method, we created all the possible environmental contact states in a non-prehensile manipulation. However, one thing that needs to be considered in non-prehensile manipulation is that the object does not make contact with the environment when it is loaded onto the mobile robot, as shown in Figure 4a. Thus, this individual state is added with the notation $C^E = \varnothing$.

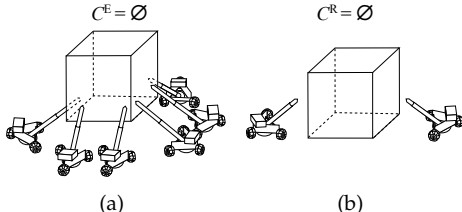

$C^E = \varnothing$　　　　　　$C^R = \varnothing$

(a)　　　　　　　　(b)

**Figure 4.** Examples of cases where (**a**) ECF is empty and (**b**) RCF is empy.

### 3.2. Generation of RCFs

RCF was easier to generate than ECF. The geometrical relationships between an object and the robots and the penetrations among them were not checked in this phase of planning because robots could locate flexibly around the targeted object, different from the case of generating the ECFs where geometrical relationships and penetrations should be checked because the geometry of the environment did not change.

When creating the possible RCF, only the robot contact on the object is considered; therefore, all the object's geometrical primitives are assumed to be accessible for robots. PCs between an object and the robots is given a definite label for each of the robots, such as $c_1^{r,ac}, c_2^{r,ac}, c_2^{r,ps}, \ldots c_n^{r,ac}$, where $n$ is the number of robots. As mentioned earlier, whether the robot always acts in passive, in active, or as switchable between passive and active depends on the functions of the robot. Then, all possible PCs between an object and the robots are combined to generate RCFs. For example, when all possible PCs between an accessible object's geometrical primitive and robots are given by $c_1^{r,ac}, c_2^{r,ps}$, the possible RCFs are obtained as $C_1^R = \{c_1^{r,ac}\}$, $C_2^R = \{c_2^{r,ps}\}$ and $C_3^R = \{c_1^{r,ac}, c_2^{r,ps}\}$ by combining the PCs.

In some states, the object lay in a stable pose, and could itself keep the contact state with the environment without any support from the robot, as shown in Figure 4b. In such cases, the RCF is an empty set given as $C^R = \varnothing$, and it is added to possible RCFs.

### 3.3. ECF-RCF Combination

With ECFs and RCFs created, the contact states of the manipulated object are generated by matching the RCFs with ECFs. In robotic manipulations, generally, force closure is required to achieve full constraints on the targeted object, which means the non-negative combination of primitive wrenches on an object can balance any external load [23]. In the case of non-prehensile manipulation, Maeda et al. [19] and Aiyama et al. [20] proposed gravity closure, which is the force closure formed by robot contacts, environmental contacts, and gravity. External loads applied on an object can be described by wrenches, and they expand the wrench vector space [24]. If sufficient wrench vector

bases are provided by the gravitational force and the contacts placed on an object, gravity closure can be achieved. However, the wrench vector bases given by contacts depend on the location and direction of the contacts determined by the configurations of the object and the robots, whereas these configurations are not concerned in the phase to generate the modes and their transitions.

Therefore, we combine an ECF and an RCF if the possible dimension of the wrench space spanned by the ECF and the RCF and the gravity satisfies the condition necessary to realize gravity closure, through which impossible modes are omitted from the viewpoint of statics.

Before calculating the dimensions of the wrench spaces spanned by the ECF and by the RCF for determining their combinations, we eliminated the infeasible combinations for a given ECF by checking whether the object's geometrical primitives were accessible to robots under any of the object's configuration. In this contact planner, we considered only the case when an object's geometrical primitives make absolute contact with the geometrical primitive of the environment; this situation would obviously disable a robot from accessing them. Therefore, for the PCs in ECF, a surface under the PC $(\text{face}, \text{face})^{\text{e}}$, an edge under the PC $(\text{edge}, \text{face})^{\text{e}}$, and a vertex under the PC $(\text{vertex}, \text{face})^{\text{e}}$ are not accessible to robots, as shown in Figure 5. The ECF that contains a certain geometrical primitive of the object that contacts the surface of the environment will not be considered to combine with an RCF that contains the same geometrical primitive of the object that contacts with robots.

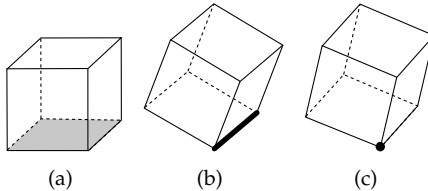

(a) (b) (c)

**Figure 5.** Three types of principal contacts (PCs) that a robot cannot access regardless of the object's configuration: (**a**) $(\text{face}, \text{face})^{\text{e}}$; (**b**) $(\text{edge}, \text{face})^{\text{e}}$; and (**c**) $(\text{vertex}, \text{face})^{\text{e}}$.

With the geometrical feasibility validated, the ECF and RCF individuals are combined according to the condition for gravity closure. Here, we considered the possible dimensions of two kind of wrench vector spaces correspondingly spanned by two kinds of forces: the passive and active forces. The passive force is applied by contacts with the environment and passive robots, and the active force is applied by contacts with the active robots and gravity. The active and passive forces were derived for the given ECFs and RCFs, respectively. Based on this, we combined the ECF and RCF, by taking into account only the derived maximum dimension of the wrench vector bases of the passive and active forces to satisfy the necessary condition of gravity closure. Configurations of the objects and the robots that meet the gravity closure will be decided in our future studies after we decide the mode transitions.

When force is applied on a contact point, the resultant wrench on the object is expressed as

$$w = \begin{bmatrix} f \\ p \times f \end{bmatrix}, \tag{1}$$

where $f$ is the force applied to the contact, and $p$ is the position of contact with respect to the object coordinate frame, as shown in Figure 6a. Certain manipulation studies have taken into account the case when a point contact also can resist a torque, which is called as a soft finger contact. However, we have not considered this type of contact in this paper.

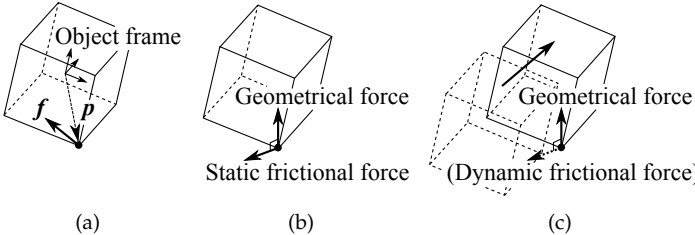

**Figure 6.** Force applied on a point of contact. (**a**) Definition of force. (**b**) Type of forces applied on a contact when the contact is fixed. (**c**) Types of forces applied on a contact when the contact is sliding.

There are two cases when a contact resists an external force: the geometrical case and the frictional case (Figure 6a,b). The former is the force caused on the object not to penetrate into the environment or the robot. The latter is the frictional force, and there are also two types of frictional forces: static and dynamic. We ignore dynamic friction when we consider the wrench bases to span gravity closure to ensure manipulation, as explained in the section of the problem statement.

For static friction on a point, the frictional cone, which is determined by the linear relationship between the geometrical force and the static frictional force, is usually considered when checking the conditions necessary for force closure. We dealt with static friction as decomposed basis independent of the geometrical force without considering the frictional cone (see Figure 7a) because our aim was to omit the modes that could not realize gravity closure regardless of the object's configuration. Therefore, the wrench space for the $j$th static or dynamic contact is expressed as

$$W_j = \langle \boldsymbol{w}_{j1}, \boldsymbol{w}_{j2}, \boldsymbol{w}_{j3} \rangle, \quad W_j = \langle \boldsymbol{w}_{j1} \rangle, \tag{2}$$

respectively, where $\boldsymbol{w}_{j1}$ is the basis for the geometrical force, and $\boldsymbol{w}_{j2}$ and $\boldsymbol{w}_{j3}$ are the bases for the static frictional force. The geometric and static frictional forces applied on edge–edge–cross contact can also be represented in the same manner (see Figure 7b). To deal with edge–face contacts and face–face contacts, we consider them as two point contacts on the edge and three non-collinear point contacts on the face (such as the boundary points of the contacting area), as shown in Figure 7c,d. The wrench space spanned by the bases is calculated by summing up the bases for all PCs in the given CF as

$$W = W_1 \cup W_2 \cup \ldots, \cup W_n, \tag{3}$$

where $n$ is the number of bases sets on the object defined in the above manner.

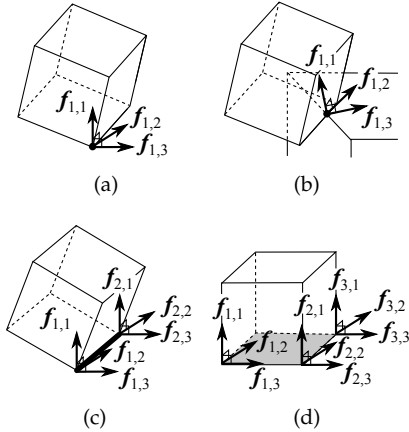

**Figure 7.** Wrench bases of the force applied on the static contact in the case of (**a**) point–face contact, (**b**) edge–face contact, (**c**) face–face contact, and (**d**) edge–edge–cross contact.

Then, the wrench spaces were separately determined for (a) the passive force applied by the contacts from the environment and the passive robots (denoted by $W^{\mathrm{ps}}$) and (b) the active force applied by gravity and the contacts from the active robots (denoted by $W^{\mathrm{ac}}$); see Figure 8. To achieve gravity closure, the wrench vectors consisting of the above forces must positively span $\mathbb{R}^6$ as follows:

$$\mathrm{pos}(W^{\mathrm{ps}} \cup W^{\mathrm{ac}}) = \mathbb{R}^6. \tag{4}$$

The reason why we separate the space into $W^{\mathrm{ps}}$ and $W^{\mathrm{ac}}$ is that the force applied by the environmental contacts and the passive robots cannot realize the force closure by themselves, and the active robots or the gravitational force must act to cause internal force on the object. According to the Carathéodory Theorem, if vectors in a wrench set positively span $\mathbb{R}^6$, the wrench set should contain at least seven vector frames in the six-dimensional space [24]. Thus, at least seven vector frames must exist in the wrench vector space consisting of forces applied by ECFs, RCFs, and gravity to achieve gravity closure. If the total dimension of the three kinds of wrench bases is less than seven, gravity closure would not be realized regardless of the configuration. Therefore, we define the necessary condition for gravity closure when combining ECFs and RCFs to omit the impossible modes as follows. Let $\dim(W^{\mathrm{ps}})$ and $\dim(W^{\mathrm{ac}})$ be the dimensions of $W^{\mathrm{ps}}$ and $W^{\mathrm{ac}}$, respectively, then an ECF and an RCF can be combined for any configuration of the object and the robots satisfying

$$\dim(W^{\mathrm{ps}}) + \dim(W^{\mathrm{ac}}) \geq 7. \tag{5}$$

To determine the modes satisfying Equation (5), we calculate the maximum dimensions of $W^{\mathrm{ps}}$ and $W^{\mathrm{ac}}$ as follows:

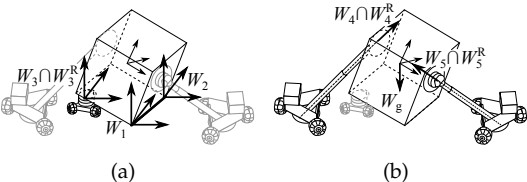

(a) (b)

**Figure 8.** Wrench spaces spanned by (**a**) forces applied by the environment–object and the passive robot–object contacts and (**b**) forces applied by the active robot–object contacts and the gravitational force.

Let there be $k$ sets of bases between an object and environment, $l$ sets of bases between an object and passive robots, and $m$ sets of bases between an object and active robots. Let $W_1$, $\ldots$, $W_{k+l+m}$ be the wrench space spanned by the contacts, which is derived from Equation (2), and the subscripts correspond with the above order. The wrench spaces simply spanned by the passive and active contacts and gravity are given by

$$W_{\mathrm{c}}^{\mathrm{ps}} = W_1 \cup \ldots \cup W_k \cup W_{k+1} \cup \ldots \cup W_{k+l} \tag{6}$$

and

$$W_{\mathrm{c}}^{\mathrm{ac}} = W_{k+l+1} \cup \ldots \cup W_{k+l+m} \cup W_{\mathrm{g}}, \tag{7}$$

respectively, where $W_{\mathrm{g}}$ is the wrench space spanned by gravity.

The wrench space spanned by RCFs requires more consideration because not only the contacts between the object and the robots, but also the kinematics of the robots affect the spanned wrench space on the object. The limitation on the available actuators of the robot, the passive joints in the robot, or other kinematics restrictions determine the possible wrench. The possible wrench space for a given robot kinematics and its configuration is also represented by the wrench bases aside from the bases of contact. For example, Figure 9 shows the wrench bases of a robot to push an object and the force bases of a robot to carry an object; these robots have passive joints and can be used in the latter section. The robot to push an object has a basis represented as a force along with the axis of

the linear actuator, as shown in Figure 9a. The robot to carry an object has three bases represented as forces passing through the axes of the passive universal joint, as shown in Figure 9b. Let $W_j^R$ represent the wrench space spanned by the robot based on its kinematics. By substituting the wrench spaces spanned by the forces applied by robot–object contacts in Equations (6) and (7) with the wrench spaces spanned by the robots based on their kinematics, the followings equations are obtained

$$W_r^{ps} = W_1 \cup \ldots \cup W_k \cup W_{k+1}^R \cup \ldots \cup W_{k+l}^R \tag{8}$$

and

$$W_r^{ac} = W_{k+l+1}^R \cup \ldots \cup W_{k+l+m}^R \cup W_g. \tag{9}$$

Taking into account the kinematics of the robots, $W^{ps}$ and $W^{ac}$ are obtained by

$$W^{ps} = W_c^{ps} \cap W_r^{ps} = W_1 \cup \ldots \cup W_k \cup [(W_{k+1} \cup \ldots \cup W_{k+l}) \cap (W_{k+1}^R \cup \ldots \cup W_{k+l}^R)] \tag{10}$$

and

$$W^{ac} = W_c^{ac} \cap W_r^{ac} = W_g \cup [(W_{k+l+1} \cup \ldots \cup W_{k+l+m}) \cap (W_{k+l+1}^R \cup \ldots \cup W_{k+l+m}^R)], \tag{11}$$

respectively. By the dimension theorem for union, $\dim(W^{ps})$ and $\dim(W^{ac})$ are given by

$$\dim(W^{ps}) = \dim(W_c^{ps}) + \dim(W_r^{ps}) - \dim(W_c^{ps} \cup W_r^{ps}) \tag{12}$$

and

$$\dim(W^{ac}) = \dim(W_c^{ac}) + \dim(W_r^{ac}) - \dim(W_c^{ac} \cup W_r^{ac}), \tag{13}$$

respectively.

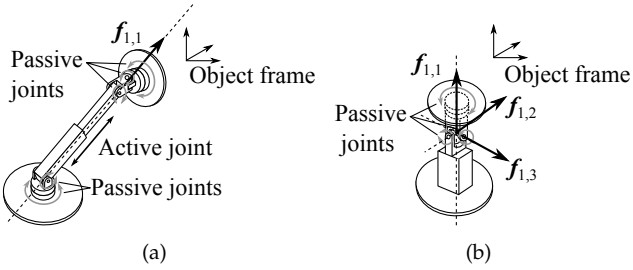

**Figure 9.** Examples having passive joints and their wrench bases. (**a**) Robot with five passive joints and a linear actuator. (**b**) Robots with three passive joints.

The maximum of $\dim(W^{ps})$ and $\dim(W^{ac})$ are difficult to derive analytically because the exact configuration of each contact state is not concerned in the current planning level. Fortunately, the maximum dimension of a contact state is a constant, except when the object lies in a singular configuration that causes dimensional degeneration. However, such kinds of singular configurations are caused only in specified configurations, such as a point or a line in the wrench space. Therefore, we sampled several configurations of objects and robots to calculate the maximum of $\dim(W^{ps})$ and $\dim(W^{ac})$. If one resultant dimension was smaller than the others, the corresponding configuration was designated as a singularity in the given ECF and RCF. The sampled $W^{ps}$ and $W^{ac}$ that provides the maximum dimension for a given ECF and RCF is considered to be the dimension of the wrench space spanned by $W^{ps}$ and $W^{ac}$.

Based on the obtained $\dim(W^{ps})$ and $\dim(W^{ac})$, ECFs and RCFs are then combined as modes if the necessary condition (i.e., Equation (5)) could be met. In this way, the modes that would appear when the robots manipulate an object are created. However, this procedure does not guarantee that the generated modes will always satisfy gravity closure, but it reduces the number of modes, as shown in latter section, by omitting the impossible modes from the viewpoint of statics.

## 4. Generation of Transitions Among Modes

In non-prehensile manipulation planning, the transition among modes is complicated because of the multiple types of contacts and their relationships. Furthermore, to obtain feasible transitions among modes, the possibility of transitions must be well reasoned by taking into account the difference between ECF and RCF in our method.

For the ECF, we consider the transition between two ECFs is determined by geometrical restrictions; therefore, we adopted the goal-contact-relaxation (GCR) graph [15,16,22] to analyze the transition among ECFs. For the RCF, because of the difference between the active robot and the passive robot, further consideration for transition is required. In this section, we propose a method to decide the transitions among modes by taking into account the following requirements for mode transition. By comprehensively considering the transition of ECF and RCF, we can obtain the mode graph, which describes the transition between the modes and represents the manipulation action sequences.

### Either ECF or RCF Changes

In non-prehensile manipulations, robots change the contacts between an object and its environment to realize the target motion. Certain manipulation planning requires that ECF and RCF changes do not occur simultaneously. Actually, simultaneous change is highly improbable in the actual manipulation. Consequently, we assume that in a mode transition process, either the RCF changes while maintaining the ECF or the RCF keeps contact to manipulate the object and to change the ECF, as shown in Figure 10a.

### Connection of ECFs in GCR Graph

The possible change of contacts between an object and the environment depends on their geometry. The object's CF can transit to its neighboring CF, which connects with it in a GCR graph [15,22]. A GCR graph is a topological graph that represents the transition among the contact states of objects. Nodes that represent the contact states of the object are connected by arcs if the corresponding states can transit to each other by compliant motions. Let $\mathcal{G}$, $V(\mathcal{G})$, and $E(\mathcal{G})$ be a GCR graph, nodes, edges in the graph, respectively. Given two primitive CFs, $\widehat{C}_a \in V(\mathcal{G})$ and $\widehat{C}_b \in V(\mathcal{G})$, $\{\widehat{C}_a, \widehat{C}_b\} \in E(\mathcal{G})$ if it is possible for the two nodes to transit to each other. Though an ECF is further split into the static and dynamic in our method, we adopted a GCR graph to judge the transition between ECFs by checking whether the set of their primitive CFs is an arc in the GCR, as shown in Figure 10b. If the corresponding primitive CFs are the same, they can transit to each other.

### Action of Active Robot

The active robot actively exerts contact forces to an object or releases contact from an object. Furthermore, the active robots changed the contact state in a system. In comparison, the passive robots could not exert contact to the object actively. They got contact with an object or released contact from an object when the object moved. Therefore, transitions that caused by the passive contact changes do not occur without the existence of active robots. Similarly, without the existence of active robots, ECFs cannot change except for the transition between isogenous ECFs, as shown in Figure 10c.

### Object Motion

As explained above, if a contact is exerted by the environment or a passive robot, the contact changes only under the object's motion. Therefore, if the object is fully constrained by the environment and the passive robots alone, the active robots cannot move the object and the transition is not caused, except for the transition between the isogenous CFs, as shown in Figure 10d.

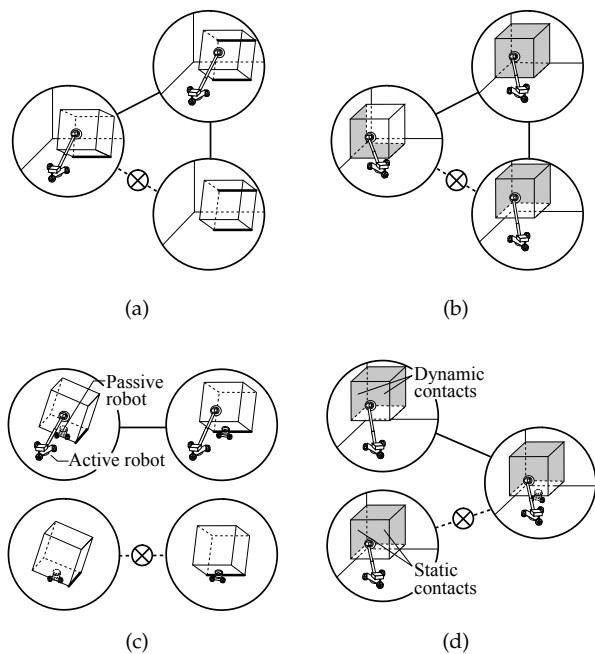

**Figure 10.** Examples of possible and impossible mode transitions. (**a**) Either ECF or RCF changes. (**b**) ECFs must be the same or connected in a goal-contact-relaxation (GCR) Graph. (**c**) Action of active robots causes a mode transition. (**d**) Object's motion caused change of ECFs and contacts of passive robots.

*4.1. Mode Graph*

With the above-mentioned requirements for transition, a mode graph, which is a graph representing the possible manipulation sequences of multiple robots, is obtained as follows. Let $\mathcal{M}$, $V(\mathcal{M})$, and $E(\mathcal{M})$ be a mode graph, nodes, and the graph edges, respectively. Let $\mathcal{G}$ be a corresponding GCR graph of ECFs with $V(\mathcal{G})$ and $E(\mathcal{G})$ as its nodes and edges, respectively. Given the two modes, $s_{\mathrm{a}} = \{C_a^{\mathrm{E}}, C_a^{\mathrm{R}}\}, s_{\mathrm{b}} = \{C_b^{\mathrm{E}}, C_b^{\mathrm{R}}\}$, where $s_{\mathrm{a}} \in V(\mathcal{M}), s_{\mathrm{b}} \in V(\mathcal{M}), \{s_{\mathrm{a}}, s_{\mathrm{b}}\} \in E(\mathcal{M})$, the following conditions are satisfied:

1.  ECF is changed: $C_a^{\mathrm{R}} = C_b^{\mathrm{R}}$ and $C_a^{\mathrm{E}} \neq C_b^{\mathrm{E}}$:

    (a)  ECFs with their primitive CFs connected in $\mathcal{G}$ are transitable under the motion of the non-fully-constrained object caused by the active robots:
    $\{\widehat{C}_a^{\mathrm{E}}, \widehat{C}_b^{\mathrm{E}}\} \in E(\mathcal{G})$, $C_a^{\mathrm{R}}$ and $C_b^{\mathrm{R}}$ include active PC, besides gravity, and $\dim(W^{\mathrm{ps}}) < 6$ in either or both $s_{\mathrm{a}}$ and $s_{\mathrm{b}}$.
    (b)  Change in isogenous contacts:
    $\widehat{C}_a^{\mathrm{E}} = \widehat{C}_b^{\mathrm{E}}$

2.  RCF is changed: $C_a^{\mathrm{R}} \neq C_b^{\mathrm{R}}$ and $C_a^{\mathrm{E}} = C_b^{\mathrm{E}}$.

    (a)  The active robot is added or removed:
    $(C_a^{\mathrm{R}} \backslash C_b^{\mathrm{R}}) \cup (C_b^{\mathrm{R}} \backslash C_a^{\mathrm{R}})$ comprises only an active PC.
    (b)  The passive robot is added or removed under the motion of a non-fully-constrained object caused by the active robots:
    $(C_a^{\mathrm{R}} \backslash C_b^{\mathrm{R}}) \cup (C_b^{\mathrm{R}} \backslash C_a^{\mathrm{R}})$ that comprise only a passive PC; $C_a^{\mathrm{R}}$ and $C_b^{\mathrm{R}}$ include an active PC, and $\dim(W^{\mathrm{ps}}) < 6$ in either or both $s_{\mathrm{a}}$ and $s_{\mathrm{b}}$.
    (c)  Change in isogenous contacts:
    $\widehat{C}_a^{\mathrm{R}} = \widehat{C}_b^{\mathrm{R}}$

### 4.2. Cost for Transition between Modes

In the generated state transition graph, the nodes were connected by arcs if the corresponding states were able to transit to each other. When we plan the manipulation based on a mode graph, we take into account the cost for transition between the nodes, which is a value defined for each arc according to the targeted manipulation task. For example, as seen in previous section, there are five kinds of connections, including transitions between the isogenous states. They may have different costs. Transitions between the isogenous states have different costs because they are just internal transitions without changes of the contact set on the object.

Given two states, we define the cost function for the transition between them as

$$l(s_a, s_b) = k, \tag{14}$$

where $k$ is the cost for $s_a$ to transform into $s_b$. In general, the manipulation path is determined by searching for path from the initial to the final states in the graph to lower the total cost. In that case, the objective function is defined as

$$\min \sum_{i=1}^{i=n-1} l(s_i, s_{i+1}), \tag{15}$$

where $s_1$ is the initial mode, and $s_n$ is the final mode.

## 5. Simulations

In this section, we showed two examples of mode generation based on the proposed method for the non-prehensile manipulation planning. We showed how the method narrows down the possible modes and their transitions and what manipulation sequences can be chosen from them based on the costs defined on the transitions.

In both examples, a certain number of robots were given, and here the important thing was that we needed to use the given robot to generate valid modes and to search for feasible manipulation paths, where the number of robots was enough to achieve the gravity closure. By searching for the paths with lowest total costs in the state transition graph, we obtained the least amount of manipulation action sequences. Since we only concerned about the transformation of contact sets in the mode transitions, we viewed the cost for each transition as the same. Therefore, we set the cost of transition between the two given modes as $k = 1$ if they contained different primitive states, either when the environmental contact or when any kind of robot contact changed. Otherwise, we set the cost as $k = 0$. Dijkstra's Algorithm was used to search for the shortest path from the initial state to the final state. We inhibited paths where the modes transited back and forth.

In the following examples, we show only the examples for generating modes and determining transitions, and we do not deal with the configuration space of an object–robot system. Therefore, although we show some figures in which the object and the robots are placed in specified configurations, they are only examples of configurations to help understated the resultant modes and transitions.

### 5.1. Example 1 and Result Discussion

In the first example, we used two types of mobile robots. The first type of robot was a pusher robot that we developed [25–27], as shown in Figure 11b. In this robot, we realized the safety manipulation that avoided the robot from falling; we did this by restricting the force that the robot could apply to the environment by using passive points. The pusher robot has a linear actuator. A face contact between the manipulator and an object acted as an active contact to change the mode of the robot–object system. The pusher robot could move to the target position by using wheels, but those wheels were lifted up and not used while the robot manipulated an object. Because of the special mechanism with passive joints, the kinematics of the robot (shown in Figure 9a) is given as explained in the previous section. See the details of the mechanism in our previous studies [25–27].

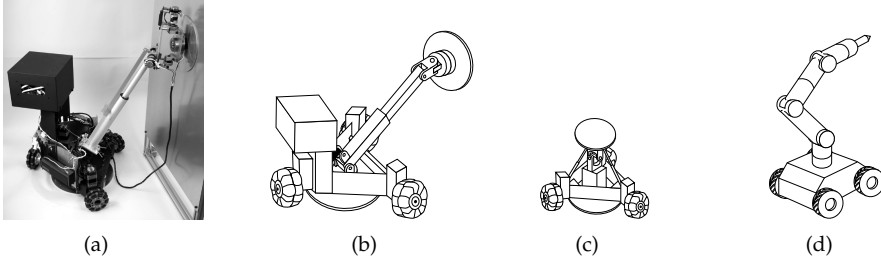

**Figure 11.** Mobile robots used in examples. (**a**) Pusher robot. (**b**) Schematic of a pusher robot. (**c**) Schematic of a transporter robot. (**d**) Mobile robot with a six-axis manipulator.

Another type of robot is a transporter robot, as shown in Figure 11c. A transporter robot (see Figure 9b) has a structure similar to the pusher robot, but it does not have a linear actuator and has fewer passive joints (as explained in the previous section). The transporter robot can also move to the target position by using wheels but those wheels are not used until it starts transporting the object. As a result, a transporter robot supports an object passively during manipulation, and a face contact between the robot and an object acted as a passive contact. We assume that a pusher robot and a transporter robot act as an active contact and a passive contact, respectively, and these functions are not changed during manipulation for simplicity.

In this manipulation, the task is to load a cuboid up to the transporter robots by using the pusher robots so that the object can be transferred away by the transporter robots. In the initial mode, the cuboid lies against a corner of the two adjacent walls, as shown in Figure 11a. The target final mode is the object held by the three transporter robots, as shown in Figure 11b. We consider the interactions between the cuboid and the environment when loading the cuboid up to the transporter robots by using the proposed method.

As shown in Figure 12, the faces, the edges, and the vertices of the cuboid are denoted as $f_1, f_2, \ldots, f_6$, $e_1, e_2, \ldots, e_{12}$, and $v_1, v_2, \ldots, v_8$, respectively. The walls are denoted as $F_1$ and $F_2$, and the floor is denoted as $F_3$. For simplicity, we assume that the frictional coefficient of the walls is small enough to deal with the contacts on them as friction-less contacts. Therefore, the ECFs including $F_1$ and $F_2$ are always dynamic contacts. We used six pusher robots whose geometrical primitives were denoted as $r_1, r_2, \ldots, r_6$ and three transporter robots whose geometrical primitives were denoted as $r_7, r_8, r_9$. For simplicity, the transporter robots and the object always made contact at the bottom of the object. Therefore, $r_7, r_8, r_9$ consisted of RCFs only with $f_2$. The initial state is given as $s_1 = \{C_1^E, C_1^R\}$, where $C_1^E = \{(f_6, F_1)^{e,dn}, (f_2, F_3)^{e,st}, (f_1, F_2)^{e,dn}\}$, and $C_1^R = \varnothing$. The final state is given as $s_G = \{C_G^E, C_G^R\}$, where $C_G^E = \varnothing$, and $C_G^R = \{(f_2, r_7)^{r,ps}, (f_2, r_8)^{r,ps}, (f_2, r_9)^{r,ps}\}$.

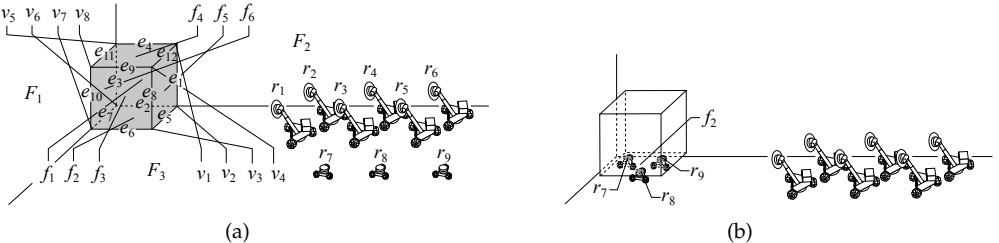

**Figure 12.** (**a**) Initial mode and (**b**) Target final mode. The targeted object for manipulation is a cuboid. There are two walls and the floor as the environment. We used six pusher robots and three transporter robots.

By applying the proposed method to generate the mode from the viewpoint of statics, 166,553,714 modes were generated, with the type and the corresponding number of robots determined according to the necessary condition to achieve gravity closure. As shown in Table 1, the total number of combinations between the possible ECFs and RCFs was 167,180,587, and we can see that proposed

protocol reduced the number of modes. The reduction in the total number of modes was not dramatical because by considering all the possible contact between the robots and the object's geometrical primitives, there existed a huge amount of RCFs. For each ECF, our method eliminated the unvailable robot combinations. However, for most of the ECFs, the environmental constraints enabled most of the robot combinations to provide sufficient constraints to achieve the gravity closure, so the reduction of the candidates of robot combination was not really much. Further, after considering the possible geometrical primitives of the object that made contact with the robots, this reduction became less significant.

The mode transition graph was generated from the above modes by using the rules defined in the previous section. Also, as shown in Table 1, the resultant number of transitions was $8.6 \times 10^9$, where we can see that the proposed rules significantly reduced the candidates of transition between modes, which originally would be $1.4 \times 10^{16}$. We can see a dramatic reduction in the number of mode transitions here because one contact states can only change to another one under the constraints of geometrical boundary relationships and under the rules that the robot contact changes one by one. With such geometrical boundary relationships and the rules to change the robot contacts considered, a mode has only very few or even no transitable modes, whereas all the other modes will be its transitable modes and a large number of meaningless transitions will be caused if our method is not adopted.

Finally, by searching the mode graph, a total of 27,216 paths which minimized the cost were obtained in the graph.

**Table 1.** Comparison among the various parameters obtained by Example 1.

|  | Before Adopting Our Method | After Adopting Our Method |
|---|---|---|
| Generated modes | 167,180,587 | 166,553,714 |
| Number of mode transitions | $1.4 \times 10^{16}$ | $8.6 \times 10^9$ |

Table 2 shows one of the obtained paths with the lowest cost in the mode graph. As mentioned above, the paths in the modes were decided without planning the exact configuration of robots. Therefore, the configurations of the object and the robots shown in Table 2 are examples of the modes. In the path shown in Table 2, the object was pushed by a pusher robot and rotated about the object's edge $e_5$ alongside the wall $F_1$. A transporter robot was then inserted underneath it. The pusher robot then moved the object so that only the object's vertex $v_3$ was in contact with the floor $F_3$, and another transporter robot was inserted underneath the object. Finally, with two transporter robots at the bottom, the object lost contact with the floor $F_3$ through contact with the pusher robot, which allowed the third transporter robot to enter underneath the object. Thus, the object was finally loaded onto three transporter robots.

We can see that the obtained path is reasonable. From the viewpoint of statics and geometry, it was also reasonable to include the paths in which only the combinations of geometrical primitives were different. The resultant paths also contained those paths that were impossible to realize from the viewpoint of the force balance determined by the configurations of the object and the robots. Those paths will be omitted in the later planning phase when deciding the object–robot system's exact configurations based on the transitions of the modes; they will not be dealt with in the current phase where only the transitions of the modes are decided. The important result is that the candidates of the mode transposition were narrowed down, as shown Table 1, and this facilitated the planning to determine the configurations.

**Table 2.** One of the manipulation paths that minimized the cost.

| Image | ECF and RCF in the Mode |
|---|---|
| | $C_1^E = \{(f_6, F_1)^{e,dn}, (f_1, F_2)^{e,dn}, (f_2, F_3)^{e,st}\}$ <br> $C_1^R = \varnothing$ |
| | $\Downarrow \quad l(s_1, s_2) = 1$ |
| | $C_2^E = \{(f_6, F_1)^{e,dn}, (f_1, F_2)^{e,dn}, (f_2, F_3)^{e,st}\}$ <br> $C_2^R = \{(f_3, r_1)^{r,ac}\}$ |
| | $\Downarrow \quad l(s_2, s_3) = 1$ |
| | $C_3^E = \{(f_1, F_2)^{e,dn}, (e_5, F_3)^{e,st}\}$ <br> $C_3^R = \{(f_3, r_1)^{r,ac}\}$ |
| | $\Downarrow \quad l(s_3, s_4) = 1$ |
| | $C_4^E = \{(f_1, F_2)^{e,dn}, (e_5, F_3)^{e,st}\}$ <br> $C_4^R = \{(f_3, r_1)^{r,ac}, (f_2, r_7)^{r,ps}\}$ |
| | $\Downarrow \quad l(s_4, s_5) = 1$ |
| | $C_5^E = \{(v_3, F_3)^{e,st}\}$ <br> $C_5^R = \{(f_3, r_1)^{r,ac}, (f_2, r_7)^{r,ps}\}$ |
| | $\Downarrow \quad l(s_5, s_6) = 1$ |
| | $C_6^E = \{(v_3, F_3)^{e,st}\}$ <br> $C_6^R = \{(f_3, r_1)^{r,ac}, (f_2, r_7)^{r,ps}, (f_2, r_8)^{r,ps}\}$ |
| | $\Downarrow \quad l(s_6, s_7) = 1$ |
| | $C_7^E = \varnothing$ <br> $C_7^R = \{(f_3, r_1)^{r,ac}, (f_2, r_7)^{r,ps}, (f_2, r_8)^{r,ps}\}$ |
| | $\Downarrow \quad l(s_7, s_8) = 1$ |
| | $C_8^E = \varnothing$ <br> $C_8^R = \{(f_3, r_1)^{r,ac}, (f_2, r_7)^{r,ps}, (f_2, r_8)^{r,ps},$ <br> $(f_2, r_9)^{r,ps}\}$ |
| | $\Downarrow \quad l(s_8, s_G) = 1$ |
| | $C_G^E = \varnothing$ <br> $C_G^R = \{(f_2, r_7)^{r,ps}, (f_2, r_8)^{r,ps}, (f_2, r_9)^{r,ps}\}$ |

## 5.2. Example 2 and Result Discussion

In the second example, a cuboid is loaded onto a step by using mobile robots, as shown in Figure 13. The faces, the edges, and the vertices of the cuboid are denoted as $f_1, f_2, \ldots, f_6, e_1, e_2, \ldots, e_{12}$, and $v_1, v_2, \ldots, v_8$, respectively. The floor is denoted as $F_1$ and the faces consisting of the step are denoted as $F_2$ and $F_3$.

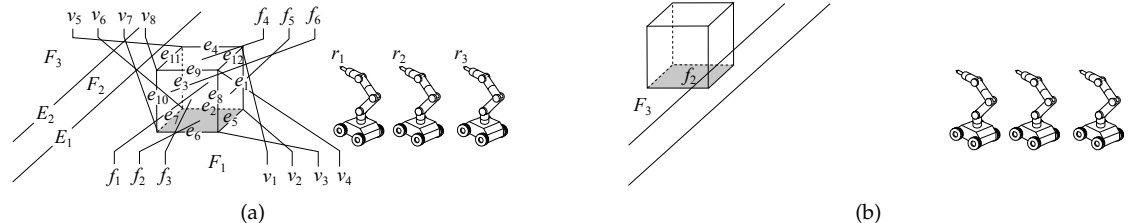

**Figure 13.** (**a**) Initial mode and (**b**) target final mode. Targeted object for manipulation is cuboid. There is a step in the environment. We used six pusher robots to bring the cuboid on the step.

A mobile robot with a standard six-axis manipulator is shown in Figure 11d. The robot generates a frictional point of contact with the object. We used three robots and denoted their geometrical primitives as $r_1, r_2$, and $r_3$. There were no obstacles around the step; therefore, the manipulation could be performed by either using or not using the environment. By using the proposed method, we generated the modes with the necessary number of robots to achieve gravity closure. Then, in the created mode graph, we took the initial state as $s_1 = \{\{(f_2, F_1)^{e,st}\}, \varnothing\}$ and the final state as $s_G = \{\{(f_2, F_3)^{e,st}\}, \varnothing\}$ and searched for the manipulation paths with the minimum total cost.

Using the procedure given in the previous example, we generated 292,415 modes and $2.9 \times 10^6$ transitions. The total number of combinations of the ECFs and RCFs was 299,967, and the total number of combinations of the generated modes was $4.5 \times 10^{10}$, as shown in Table 3. For the same reason we analyzed in Example 1, our method also narrowed down the number of possible modes, and especially, dramatically decreased the transitions between them.

**Table 3.** Comparison among the various parameters obtained by Example 2.

|  | **Before Adopting Our Method** | **After Adopting Our Method** |
|---|---|---|
| Generated modes | 299,967 | 292,415 |
| Number of mode transitions | $4.5 \times 10^{10}$ | $2.9 \times 10^6$ |

Our search produced 672,352 paths with a minimum total cost. In these obtained paths, the number of robots involved varied. In the two paths shown in Table 4, we found that the number of robots engaged in the two paths were different.

In Path 1, only two robots were needed to perform the manipulation task whereas three robots were needed in Path 2 because the floor was used to exert constraints on the object. Therefore, fewer robots were needed in Path 1. We could choose the path according to some other criteria based on the obtained paths. For example, the manipulation action sequences obtained from Path 1 would be more preferable than those obtained from Path 2 because fewer robots were adopted. Besides, as compared with the previous example, relative sliding occured on the face contact principals in the action sequences of Path 1 between the mode $s_5 = \{\{(e_5, F_1)^{e,dn}\}, \{(f_3, r_1)^{r,ac}, (f_5, r_2)^{r,ac}\}\}$ and the mode $s_6 = \{\{(e_5, F_1)^{e,dn}, (f_2, E_2)^{e,dn}\}, \{(f_3, r_1)^{r,ac}, (f_5, r_2)^{r,ac}\}\}$; therefore, the transition from static contact to dynamic contact appeared between the mode $s_4 = \{\{(e_5, F_1)^{e,st}\}, \{(f_3, r_1)^{r,ac}, (f_5, r_2)^{r,ac}\}\}$ and $s_5$. A similar transition can also be seen in Path 2.

In the non-prehensile manipulation by multiple mobile robots, there were many possible manipulation sequences each with a difference number of robots. As shown in above example, our method can list up possible patterns of manipulation taking into account their possibility from the viewpoint of statics.

**Table 4.** Two paths of the manipulation paths which minimized the cost.

| Path 1 | | Path 2 | |
|---|---|---|---|
| **Image** | **ECF and RCF in the Mode** | **Image** | **ECF and RCF in the Mode** |
|  | $C_1^E = \{(f_2, F_1)^{e,st}\}$ <br> $C_1^R = \varnothing$ |  | $C_1^E = \{(f_2, F_1)^{e,st}\}$ <br> $C_1^R = \varnothing$ |
| | $\Downarrow \quad l(s_1, s_2) = 1$ | | $\Downarrow \quad l(s_1, s_2) = 1$ |
|  | $C_2^E = \{(f_2, F_1)^{e,st}\}$ <br> $C_2^R = \{(f_3, r_1)^{r,ac}\}$ |  | $C_2^E = \{(f_2, F_1)^{e,st}\}$ <br> $C_2^R = \{(f_3, r_1)^{r,ac}\}$ |
| | $\Downarrow \quad l(s_2, s_3) = 1$ | | $\Downarrow \quad l(s_2, s_3) = 1$ |
|  | $C_3^E = \{(e_5, F_1)^{e,st}\}$ <br> $C_3^R = \{(f_3, r_1)^{r,ac}\}$ |  | $C_3^E = \{(f_2, F_1)^{e,st}\}$ <br> $C_3^R = \{(f_3, r_1)^{r,ac}, (f_5, r_2)^{r,ac}\}$ |
| | $\Downarrow \quad l(s_3, s_4) = 1$ | | $\Downarrow \quad l(s_3, s_4) = 1$ |
|  | $C_4^E = \{(e_5, F_1)^{e,st}\}$ <br> $C_4^R = \{(f_3, r_1)^{r,ac}, (f_5, r_2)^{r,ac}\}$ |  | $C_4^E = \{(f_2, F_1)^{e,st}\}$ <br> $C_4^R = \{(f_3, r_1)^{r,ac}, (f_5, r_2)^{r,ac}, (f_1, r_3)^{r,ac}\}$ |
| | $\Downarrow \quad l(s_4, s_5) = 0$ | | $\Downarrow \quad l(s_4, s_5) = 1$ |
|  | $C_5^E = \{(e_5, F_1)^{e,dn}\}$ <br> $C_5^R = \{(f_3, r_1)^{r,ac}, (f_5, r_2)^{r,ac}\}$ |  | $C_5^E = \varnothing$ <br> $C_5^R = \{(f_3, r_1)^{r,ac}, (f_5, r_2)^{r,ac}, (f_1, r_3)^{r,ac}\}$ |
| | $\Downarrow \quad l(s_5, s_6) = 1$ | | $\Downarrow \quad l(s_5, s_6) = 1$ |
|  | $C_6^E = \{(e_5, F_1)^{e,dn}, (f_2, E_2)^{e,dn}\}$ <br> $C_6^R = \{(f_3, r_1)^{r,ac}, (f_5, r_2)^{r,ac}\}$ |  | $C_6^E = \{(f_2, F_3)^{e,dn}\}$ <br> $C_6^R = \{(f_3, r_1)^{r,ac}, (f_5, r_2)^{r,ac}, (f_1, r_3)^{r,ac}\}$ |
| | $\Downarrow \quad l(s_6, s_7) = 1$ | | $\Downarrow \quad l(s_6, s_7) = 0$ |
|  | $C_7^E = \{(f_2, E_2)^{e,dn}\}$ <br> $C_7^R = \{(f_3, r_1)^{r,ac}, (f_5, r_2)^{r,ac}\}$ |  | $C_7^E = \{(f_2, F_3)^{e,st}\}$ <br> $C_7^R = \{(f_3, r_1)^{r,ac}, (f_5, r_2)^{r,ac}, (f_1, r_3)^{r,ac}\}$ |
| | $\Downarrow \quad l(s_7, s_8) = 1$ | | $\Downarrow \quad l(s_7, s_8) = 1$ |
|  | $C_8^E = \{(f_2, F_3)^{e,st}\}$ <br> $C_8^R = \{(f_3, r_1)^{r,ac}, (f_5, r_2)^{r,ac}\}$ |  | $C_8^E = \{(f_2, F_3)^{e,st}\}$ <br> $C_8^R = \{(f_3, r_1)^{r,ac}, (f_5, r_2)^{r,ac}\}$ |
| | $\Downarrow \quad l(s_8, s_9) = 1$ | | $\Downarrow \quad l(s_8, s_9) = 1$ |
|  | $C_9^E = \{(f_2, F_3)^{e,st}\}$ <br> $C_9^R = \{(f_3, r_1)^{r,ac}\}$ |  | $C_9^E = \{(f_2, F_3)^{e,st}\}$ <br> $C_9^R = \{(f_3, r_1)^{r,ac}\}$ |
| | $\Downarrow \quad l(s_9, s_G) = 1$ | | $\Downarrow \quad l(s_9, s_G) = 1$ |
|  | $C_G^E = \{(f_2, F_3)^{e,st}\}$ <br> $C_G^R = \varnothing$ |  | $C_G^E = \{(f_2, F_3)^{e,st}\}$ <br> $C_G^R = \varnothing$ |

## 6. Conclusions

In this paper, we proposed the modal planning method for multi-contact non-prehensile manipulation using multiple mobile robots. After defining a mode as a set of configurations that hold the same contact state and investigating the transition between modes, a manipulation planner can efficiently probe the configuration space even if the states under varying constraints are difficult to sample in the configuration space. When multiple mobile robots manipulate the object using non-prehensile methods, the modes and their consequent transitions become enormous because of the numerous contacts made by the environment and the robots on the object. For such situations, we proposed a method to narrow down the possible modes and their transitions beforehand by excluding the invalid modes and transitions. In our proposed method, we generated modes that

described an object's contact states with the robots and the environment while ignoring their exact configurations, provided each multi-contact set satisfied the necessary condition for gravity closure on the object (along with gravity). Secondly, we investigated the valid transition between the modes by taking into account whether the given robot could actively change an object's contact state under feasible geometrical relationships. Finally, we conducted two simulations on specific manipulation tasks to validate our method and confirmed that the number of modes and transitions had significantly reduced. Also, it was feasible to obtain the sequence of modes obtained by searching the shortest path.

Our method can be adopted to probe the modal spaces in the variant cases of non-prehensile manipulation by mobile robots. If prior sequencing manipulation actions by adopting the generated mode transition graph, the manipulation planner can avoid the heavy computation of searching in a whole large configuration space. Thus, determining the sequence of modes is usually the first hierarchy in manipulation planning, where we do not consider the exact configuration of a system that comprises objects and robots. However, these configurations must be determined to complete manipulation planning by considering certain factors, such as the achievement force closure, the movability of the object, and accessibility for mobile robots. We can obtain those configurations based on the prior determined modal space, by applying sample-based methods to each modal space. This will be significantly more efficient than directly applying sample-based methods to probe a whole configuration space. In our future studies, we propose to further investigate this topic.

**Author Contributions:** Conceptualization, J.O., S.S. and C.F.; Methodology, C.F., S.S. and J.O.; Software, C.F. and S.S.; Validation, C.F. and S.S.; Data curation, C.F. and S.S.; Formal analysis, C.F., S.S. and J.O.; Investigation, C.F., S.S. and J.O.; Resources, J.O.; Writing–original draft preparation, C.F. and S.S.; Writing–review and editing, C.F., S.S. and J.O.; Visualization, S.S. and C.F.; Supervision, S.S. and J.O.; Project administration, J.O.; Funding acquisition, J.O.

**Funding:** This research received no external funding.

**Acknowledgments:** Changxiang Fan is financially supported by China Scholarship Council (CSC).

**Conflicts of Interest:** The authors declare no conflict of interest.

## Abbreviations

The following abbreviations are used in this manuscript:

PC　　　principal contact
CF　　　contact formation
ECF　　　environmental contact formation
RCF　　　robot contact formation
GCR　　　goal-contact-relaxation

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
