# Peer review of "Modal Planning for Cooperative Non-Prehensile Manipulation by Mobile Robots"

_applsci, doi:10.3390/app9030462_

Round 1

Reviewer 1 Report

Dear authors,

The paper presents a modal planning method for non-prehensile manipulation using multiple mobile robots. The paper includes an extensive introduction and the problem is described in detail. It also includes a huge explanation of the modes generation and two simulations that validate the proposed method. 

I have some short comments about the paper:

- In line 131 the first reference to Figure 3 is made. However, up to this point there is no explanation of the ECF, RCF abbreviations. This is then indicated in lines 151. Maybe all these terms should be explained before.

- I don't know if the number of robots is limited in some way in your method. Table 4 shows two paths using different number of robots. Is this number limited? How does the method determine the number of robots? Have all the robots the same "weight" ?Maybe it could be helpful to include an explanation.

I would thank you if these suggestions could be answered.

Best regards

Author Response

Dear Reviewer,

Thank you very much for your fruitful comments.

For your first comment:

- In line 131 the first reference to Figure 3 is made. However, up to this point there is no explanation of the ECF, RCF abbreviations. This is then indicated in lines 151. Maybe all these terms should be explained before.

Thank you for pointing out this deficiency in our paper. After we considered the content again, we found that the reference to Figure 3 in line 131 seems not introducing the corresponding content clearly, which also happened to line 142 and 144. To avoid making the readers confused, we remove the reference to Figure 3 that appeared in line 131, 142 and 144. After this revision, now we refer to the Figure 3 only when we propose the definition of environmental contact formation (ECF) and robot contact formation (RCF).

For your second comment:

- I don't know if the number of robots is limited in some way in your method. Table 4 shows two paths using different number of robots. Is this number limited? How does the method determine the number of robots? Have all the robots the same "weight"? Maybe it could be helpful to include an explanation.

In general cases of our method, there is no maximum limitation to the number of robots, but there is minimum limitation to it in each mode for the sake of manipulation stability, and that is why we determine the combination of ECF and RCF by investigating whether the contact set can form the gravity closure.

However, in Simulation 2, which is related to Table 4, we limited the number of robots, because only 3 robots were provided (it was also the same case in Simulation 1). We believe such cases also happen in other similar manipulation cases that only certain number of robots are given. We show in Table 4 two paths using different number of robots, because in our result we found even in the shortest paths with the same amount of action sequences, the engaged number of robots would be different. In the case when we want to use fewer robot to finish the manipulation task (or when any of the given robot meets faults), the paths with fewer number of robots would be preferable to do the further configuration planning for the system.

To determine the number of robots, firstly we generated the modes with enough number of robots for manipulation stability, by investigating whether the environmental contact, robot contact and the object’s gravity can together achieve the gravity closure. We also considered the robot contact only change one by one in each mode transition. In this way, in the process of mode transition from the initial mode to the targeted mode, we always have enough number of robots to transform the current stable mode to the next one. Thus, the engaged number of robots in this manipulation path is the number of robots determined by our method.

In our simulation, all robot had the same “weight”, which we described by mentioning the cost to be 1 in each mode transition. Because we only concerned about the transformation of contact state in our research. If the contact set on the object changed, either environmental contact or any kind of robot contact, the cost for the corresponding mode transition was 1. We are sorry that we did not explain clearly about this part, and we already made some revision in the second paragraph the section Simulations our paper as:

In both examples, a certain number of robots were given, and here the important thing was that we needed to use the given robot to generate valid modes and to search for feasible manipulation paths, where the number of robots was enough to achieve the gravity closure. By searching for the paths with lowest total costs in the state transition graph, we obtained the least amount of manipulation action sequences. Since we only concerned about the transformation of contact sets in the mode transitions, we viewed the cost for each transition as the same. Therefore, we set the cost of transition between the two given modes as k=1 if they contained different primitive states, either when the environmental contact or when any kind of robot contact changed. Otherwise, we set the cost as k=0. Dijkstra's Algorithm was used to search for the shortest path from the initial state to the final state. We inhibited paths where the modes transited back and forth.

Further corrections have been made based on the comments of another reviewer.

Again, thank you very much for your precise comments.

Reviewer 2 Report

The paper presents a method of mode planning for non-prehensile manipulation by using mobile robots. This method can reduce the possible modes and transitions by considering the statics, kinematics and geometry. The paper describes the definitions of the modes, transitions, and how to use such method by showing two examples.   

The paper is well organized and clear. The results seem very interest for the aim and scope of the Applied Sciences. In my opinion, the paper is accepted in present form.

Author Response

Dear Reviewer,

Thank you very much for taking your time to read our paper and give us your fruitful comments.

Your comments support and affirm the value of our research. Aimed at a furtherly developed method, we are proceeding our research and hoping to show a better result in the following work of this paper. We still noticed some deficiencies in our paper and have made some revisions to our paper. Corrections have also been made based on the comments of other reviewers. We have uploaded our revised version of the paper. We appreciate your further comments if you have time to read the updated version.

Again, thank you very much for your precise comments.

Reviewer 3 Report

The paper presents a method to reduce optimize the modal planning for cooperative non-prehensile manpulation using mobile robots.

The study about the state of the art is correct and it is also well introduced in the paper. The authors made a correct use of current technological background to create a soundness method to reduce the number of modes and transitions during planning.

Results and analysis is adecuated for a theoretical study.

Author Response

(The authors gave the same response as above.)
